# A Pilot Study of Bioenergetic Marker Relationships in Gulf War Illness: Phosphocreatine Recovery vs. Citric Acid Cycle Intermediates

**DOI:** 10.3390/ijerph18041635

**Published:** 2021-02-09

**Authors:** Beatrice A. Golomb, Hayley J. Koslik, Jun Hee Han, Anna Helena Preger Guida, Gavin Hamilton, Richard I. Kelley

**Affiliations:** 1Department of Medicine, School of Medicine, University of California, San Diego, CA 92093-0995, USA; hkoslik@idcrp.org (H.J.K.); jhh006@ucsd.edu (J.H.H.); apregerg@ucsd.edu (A.H.P.G.); 2Department of Radiology, University of California, San Diego, CA 92093-0995, USA; ghamilton@health.ucsd.edu; 3Department of Genetics and Genomics, Boston Children’s Hospital, Boston, MA 02115, USA; rikelleymd@comcast.net

**Keywords:** Gulf War illness, veterans, bioenergetics, citric acid cycle, 31-phosphorus magnetic resonance spectroscopy

## Abstract

Impaired bioenergetics have been reported in veterans with Gulf War illness (VGWIs), including prolonged post-exercise recovery of phosphocreatine (PCr-R) assessed with ^31^Phosphorus magnetic resonance spectroscopy. The citric acid cycle (CAC) is considered the most important metabolic pathway for supplying energy, with relationships among CAC markers reported to shift in some but not all impaired bioenergetic settings. We sought to assess relations of CAC markers to one another and to PCr-R. Participants were 33 VGWIs and 33 healthy controls 1:1 matched on age–sex–ethnicity. We assessed seven CAC intermediates, and evaluated PCr-R in a subset of matched case–control pairs (N = 14). CAC markers did not significantly differ between cases and controls. Relationships of alpha-ketoglutarate to malate, isocitrate, and succinate were strongly significant in cases with materially weaker relationships in controls, suggesting possible shifts in these markers in concert in VGWIs. PCr-R correlated strongly with five of seven CAC markers in controls (succinate, malate, fumarate, citrate, isocitrate, range r = −0.74 to −0.88), but bore no relationship in VGWIs. In summary, PCr-R related significantly to CAC markers in healthy controls, but not VGWIs. In contrast, relations of CAC markers to one another appeared to shift (often strengthen) in VGWIs.

## 1. Introduction

Gulf War illness (GWI) is a chronic multi-symptom condition that affects an estimated one third of the ~700,000 US personnel deployed to the 1990–1991 Persian Gulf theater. The ground war lasted only four days, many of those deployed never saw combat, and combat stress is not an independent predictor of GWI [1,2]. However, this conflict involved a large number of new, unique, and excessive environmental and drug exposures [3]. For instance, these included organophosphates as nerve gas for the estimated 100,000 exposed following the demolition of the Khamisiyah weapons depot; organophosphate, carbamate, and other pesticides; pyrethroid-impregnated uniforms; pyridostigmine bromide (a carbamate) as a nerve agent pretreatment adjunct; the first widespread US military use of anthrax vaccine; the only military use of botulinum toxoid vaccine; and the first use of depleted uranium (to gird tanks so that they could not be penetrated by enemy munitions, and to strengthen munitions to better penetrate enemy tanks); as well as oil fires, burn pits, and other exposures. Studies tie environmental and drug exposures to GWI (and its defining symptoms) [1,4]; tie exposures to mitochondrial impairment [5,6]; and tie mitochondrial impairment to GWI (and GWI symptoms) [7,8,9,10].

Bioenergetic impairment has been shown in veterans with GWI (VGWIs). For instance, post-exercise phosphocreatine recovery time constant (PCr-R) upon ^31^Phosphorus magnetic resonance spectroscopy (^31^P-MRS), was significantly prolonged in VGWIs (selected by Kansas and CDC GWI criteria) [7]. Mitochondrial/bioenergetic impairment is known to produce multi-symptom illness such as that reported in VGWIs, with an emphasis on fatigue, brain, and muscle symptoms (as observed in VGWIs)—differentially producing symptoms in post-mitotic organs of high energy demand [11]. This index has been deemed a robust index of mitochondrial impairment in vivo [12]. For this purpose, mitochondrial impairment can be construed broadly to encompass reduced energy supplied by mitochondria for reasons that may extend beyond respiratory chain dysfunction per se. These can include, for example, reduced mitochondrial number, transport, impaired energy production by mitochondria due to substrate or oxygen availability limitations, etc. Another study reported increased mitochondrial DNA lesion frequency and mitochondrial DNA copy number in peripheral blood mononuclear cells of VGWIs relative to healthy controls [10]. Of note, alterations in mitochondrial lipids and function have been shown in animal models of GWI [13,14,15].

There is need to assess additional markers that bear a potential relation to bioenergetics, to begin to better understand the (range of) contributors to GWI, and the interrelation among such markers, including whether or how these relationships are altered in GWI. We therefore sought to assess concentrations of citric acid cycle (CAC) intermediates in VGWIs and in matched healthy controls. “The citric acid cycle is the final common oxidative pathway for carbohydrates, fats and amino acids. It is the most important metabolic pathway for the energy supply to the body” [16]. In fact, though it is often depicted as a unidirectional cycle, most of the reactions are bidirectional—all except citrate synthase and alpha-ketoglutarate dehydrogenase [17]—so influences can occur at the level of individual CAC elements. Although CAC marker levels are not good indices of energetic status, shifts in how CAC markers relate to one another have been reported in settings of impaired energy/altered bioenergetics [17,18,19]. For instance, in one paper examining citric acid cycle intermediates in infants with mitochondrial disorders, it is stated: “Most of the Krebs Cycle intermediates were also not useful in discriminating patients with mitochondrial disorders. Interestingly, there was strikingly poor correlation among most of those analytes in all patient groups, but fumarate and malate were uniquely well correlated (r^2^ = 0.840)” [18]. We sought to assess the hypothesis that CAC markers’ relation to one another, and to PCr-R, was shifted in VGWIs relative to healthy controls.

## 2. Materials and Methods

### Design: Case–Control Study

Participants: 33 healthy controls were 1:1 matched to 33 VGWIs. Participants were 33 matched pairs from a study on which prostaglandin/leukotriene results have been published [20]. ^31^P-MRS was secured from a sub-study, under separate funding (for which an IRB amendment to the parent study was secured), in which participants were a proper subset of participants in the study from which CAC markers were procured.

Recruitment: Our procedures for recruitment of VGWIs have been previously reported: “Categories of recruitment approach included directed as well as general media, collaborations with support groups/interest groups, local free advertising resources (Craigslist and Backpage), physician outreach, Internet-based approaches, and referrals from study participants and screenees” [21]. The present study capitalized on past participants that had been recruited via the above approach. For controls, recruitment modalities included the above approaches, as well as ResearchMatch [22] and postings at our university’s clinical research center.

Ethics statement: Approval for the parent study and for the ^31^P-MRS sub-study were both secured on 5 July 2010, under the UCSD Human Research Protections Program (HRPP) #100959. All participants gave written informed consent to participate.

Cases: To qualify as a VGWI, veterans must have been deployed to the Gulf theater at any time between August 1990 and July 1991. They additionally had to meet the Centers for Disease Control and Prevention (CDC) and Kansas symptom inclusion criteria for GWI [23,24]. CDC criteria require symptoms for at least 6 months arising since the advent of the Gulf War in at least two of three domains of fatigue/sleep, mood/cognitive, and musculoskeletal symptoms [23]. The more specific and discriminating Kansas criteria require symptoms for at least 6 months arising since the advent of the Gulf War in at least 3 of 6 categories of fatigue/sleep, pain, neurological/cognitive/mood, respiratory, gastrointestinal, and dermatologic symptoms [24]. Moreover, to qualify in a category, the symptoms must be at least moderate in severity (not mild) and/or there must be multiple symptoms within the category. Kansas criteria request that designated component symptoms that qualify for a domain be self-rated from 0–3 (as absent, mild, moderate, severe). Participants could not have conditions, such as lupus or multiple sclerosis, that produce chronic symptoms that could be confused for GWI.

Controls: To qualify as a control, prospective participants were healthy nonveterans meeting neither Kansas nor CDC symptom inclusion criteria, and without Kansas exclusion criteria (that is, they could not have other health conditions such as lupus or multiple sclerosis that could produce symptoms that might be described similarly to those of GWI). Controls were matched 1:1 to cases on sex, ethnicity, and age (within four years). In recognition of the fact that mixed ethnicities are common, a half-match for ethnicity was qualifying.

Measurements: (a) Survey measures: In addition to surveys assessing eligibility criteria, which included symptoms, participants completed surveys related to demographics, as well as Gulf and non-Gulf exposures. (b) CAC markers: CAC intermediates were assessed in plasma samples of participants, by coauthor Kelley, using ethylacetate extraction and gas chromatography/mass spectrometry, according to the methods of Sandlers et al. [25]. Assessments were conducted without access to case vs. control information. CAC measurements were conducted in all 66 participants (33 matched pairs). However, due to a laboratory issue affecting samples for two case–control pairs, citrate and isocitrate were each assessed in 31 matched pairs. (c) PCr-R: Phosphocreatine post-exercise recovery time constant was assessed (as previously reported) in 14 of the participants (7 GWI cases and their 7 matched controls) by ^31^P-MRS [7]. Briefly, using a 3T scanner and 5-inch diameter surface coil under the calf, spectra were assessed every 3 s over 2 min of rest, 5 min of calf exercise (foot pedal depression exercise), then 6 min of recovery. This assessed phosphorus-containing products including phosphocreatine, a backup energy source for muscle that is depleted during exercise, with recovery rate dependent on ATP generation [12]. An exponential was fitted to determine PCr-R. As above, this assessment has been deemed a robust index of mitochondrial function in vivo [12].

Of note, CAC markers involved 1:1 paired cases and controls. PCr-R involved 1:1 paired cases and controls, and cases and controls were a subset of those in the CAC marker group; but the 1:1 pairings were not all shared across the two sets.

Analysis: Descriptive statistics characterized all 66 participants (cases and controls), and the subset of 14 in whom PCr-R was available from ^31^P-MRS. *T*-tests for continuous markers, and chi-square tests for categorical markers, compared characteristics in cases to those in their matched controls (paired *t*-test). Paired *t*-tests assessed the relationship of CAC markers to case–control status. Because of differences in time from visit date to the mailing and testing of archive samples, for each CAC marker, we assessed whether there was a relation to archive time, separately in cases and controls, using regression with robust standard errors. Correlation was used to assess relationships among CAC markers, separately for cases and controls. For any CAC marker with a suggestion of a relation to archive time, relationships were reappraised using regression adjusted for archive time. All regression analyses employed robust (heteroskedasticity-independent) standard errors. Correlations of each CAC marker to PCr-R, and regression with robust standard errors predicting each CAC marker by PCr-R (adjusting for archive time, for CAC markers with any possible relation to archive time), were analyzed separately for cases and controls. Analyses were carried out using Stata™ version 9.0 and 12.0 (StataCorp LLC, College Station, USA). Two-sided *p*-values < 0.05 determined statistical significance.

## 3. Results

Characteristics of cases and controls are shown in Table 1a (the full 66, 33 each of cases and controls) and Table 1b (the 14 for whom ^31^P-MRS was undertaken). Cases and controls were, by selection, similar in age, sex, and ethnicity. As has been reported in other studies, VGWIs were significantly more likely to be married than were controls, possibly reflecting that for veterans with these challenging conditions, it is primarily those with social support that can add study participation to already challenging lives [20].

In contrast to PCr-R, values of CAC markers did not significantly differ between cases and controls, in either the total sample or the subsample in whom PCr-R was measured (Table 2a,b).

The relation of CAC markers to visit date was assessed (the relationship is equivalent to that for archive time except for the constant), and was assessed separately in all cases and all controls. There was a suggestion of a relationship for two CAC markers, alpha ketoglutarate (AKG) and aconitate. In controls (regression with robust SE), the relationships, β (SE) p, were AKG: +0.019 (0.011) *p* = 0.10; aconitate: −0.0016 (0.00092), *p* = 0.093. In cases, AKG: +0.0073 (SE 0.0097), *p* = 0.46; aconitate: −0.16 (SE 0.00090), *p* = 0.083. For no other CAC marker was the relationship near significant in either group (e.g., no *p* < 0.25 in either group).

Table 3a,b show correlations of CAC markers to one another, separately in cases and controls. All correlation coefficients were positive in each group. Note some strong relationships (correlations ~0.5 or greater) were shared between cases and controls with a focus on correlations. Other correlations were strong (r > 0.5) in either cases or controls, and materially weaker in the other group. See, for instance, the AKG relation to malate and to isocitrate; and the isocitrate relation to fumarate. Citrate and isocitrate related strongly to fumarate, in controls but not in cases. There was a less striking difference in correlation for citrate vs. succinate (r = 0.39 vs. r = 0.56).

Table 4 shows results for AKG and aconitate. Samples were stored at −80 °C prior to transport to the laboratory for CAC intermediate testing. They were shipped overnight on dry ice on a weekday before a Wednesday, to ensure receipt prior to the weekend. Since archive time appeared to affect these variables, relationships were evaluated while adjusting for the time variable. Using regression with robust SEs, the time-affected CAC marker was the designated outcome variable, with the other CAC marker as the predictor adjusting for archive time. In archive time-adjusted analysis (but not unadjusted), the AKG and aconitate relation to succinate was highly significant in cases but not controls. AKG related strongly to isocitrate and to malate in cases but not controls (the magnitude of these differences was large). In contrast, aconitate related strongly to malate in controls but not cases.

Table 5 and Appendix A show the relationship of CAC markers to PCr-R in controls and GWI cases. In controls, all correlations of CAC markers to PCr-R share a common sign (negative, as might be expected if prolonged PCr-R is indexing a shortfall in energetics of a character that is reflected in lower CAC markers). The typical magnitude of the correlations (absolute value of coefficient) was large: five of these CAC correlations to PCr-R in controls bore a correlation coefficient (absolute value) of 0.74 or greater (all except AKG and aconitate and the lowest absolute magnitude correlation, for aconitate, was 0.34). Correlations of PCr-R to all CAC markers except AKG and aconitate (the two CAC markers influenced by archive time) were significant or borderline significant in controls.

In contrast, for GWI cases, correlations of PCr-R to CAC markers were inconsistent in sign, low in absolute magnitude, and nonsignificant in all instances (no *p* < 0.5—very far from *p* < 0.05). Four correlation coefficients had a positive sign, and three had negative sign. Absolute values of all coefficients were less than the absolute value of any coefficient among controls (maximum case coefficient 0.29 vs. control 0.88). Thus, among GWI cases, the apparent potent relationship between CAC marker values and PCr-R was lost.

## 4. Discussion

### 4.1. Recap of Findings

Though previous studies have shown evidence of bioenergetic impairment in VGWIs, CAC intermediates did not differ in mean value between cases and controls, including between cases and controls who differed in PCr-R, an index of bioenergetic status. However, the relation of CAC markers to one another was materially shifted in VGWIs relative to controls. In both groups, all correlations of CAC markers to one another were positive; however, AKG relationships to malate and isocitrate, and possibly succinate, were strong and highly significant in cases, and materially weaker in controls. Conversely, relationships such as those of fumarate to citrate and to isocitrate, though strong and very highly significant in controls, were markedly weaker and nonsignificant in cases.

Yet more striking differences were observed in relations of CAC markers to PCr-R when considering an index of mitochondrial/bioenergetic status, which had been assessed in a subset using ^31^P-MRS. Higher CAC marker values in controls related significantly to faster PCr-R in controls for five of the seven CAC markers (all but the two bearing a relation to archive time), with strong correlation coefficients, from −0.74 to −0.88. This is consistent with prolonged PCr-R serving to index impaired energy production, also reflected in low levels of mitochondrion-derived CAC markers (in those controls with more prolonged PCr-R). In contrast, PCr-R bore *no trend of a relationship to any* CAC markers in GWI cases and lost the consistent direction (sign) of the relationship. Four of the seven correlation coefficients were positive (the consistent direction of the relationship was lost), and no relationship had a *p*-value < 0.5. Thus, the strong and significant relationships with PCr-R spanning multiple CAC markers observed in controls were absent in cases.

### 4.2. Fit with Existing Literature

We are aware of no other study that has assessed CAC intermediates and PCr-R in individuals with or without mitochondrial impairment; our findings on this point are entirely novel.

Altered relationships among CAC markers in settings of bioenergetic dysregulation have been previously reported [17,18,19]. However, consistent with our failure to find significant case–control differences, it was elsewhere observed that CAC intermediates are in general not useful in diagnosing mitochondrial disorders [18]. The authors noted that “Interestingly, there was strikingly poor correlation among most of those analytes in all patient groups” [18] (in their sample, fumarate and malate appeared to be well correlated).

While values were not significantly elevated in VGWIs relative to controls, AKG bore an apparently strengthened relationship to isocitrate, malate, and (in the time-adjusted analysis only) succinate in cases, suggesting a possible shift in concert in cases relative to controls. In the study that reported the above finding, it was expressly noted that citrate was not elevated in parallel with CAC markers that were raised; it may merit note that relationships of citrate to other CAC intermediates that were observed in controls were lost or markedly attenuated in cases.

One study looked at settings of anion gap metabolic acidosis, associated with elevated d-lactate, and consistent with energy dysregulation [19]. VGWIs are neither known to have, nor to lack, higher typical anion gaps, but our point is to look at settings in which relationships among CAC markers are altered. In these patients, AKG, isocitrate, and malate were found to be elevated, and in the subset without known cause for their anion gap acidosis, succinate was elevated with them [19]. It has elsewhere been reported that in anaerobic states, AKG can backflux to isocitrate, that NAD^+^ will originate from malate dehydrogenase, “operating toward malate formation”, and that “under anaerobic conditions, succinate formation is favored” [17]. These factors suggest that in some settings, impaired energy adequacy may drive AKG, isocitrate, malate, and succinate in a common direction. Such a phenomenon could be hypothesized to contribute to enhanced correlations among these markers, observed in GWI cases.

### 4.3. Limitations

VGWIs were more likely to be married than controls. As has been noted elsewhere, “GWI is a demanding condition, and those who have the social support of marriage may be those most able to incorporate research participation into their difficult lives” [20]. However, there is no a priori basis for presuming that marital status will be tied to either CAC intermediates or PCr-R, to influence differences, where present, between the groups.

Most participants were male, ^31^P-MRS participants were exclusively male, and GWI findings have repeatedly been reported to differ in male versus female veterans, and findings may not necessarily generalize to female veterans.

CAC marker assessments were conducted from plasma. In general, with mitochondrial assessments, it is desirable to sample affected tissue. It would be desirable to reappraise CAC intermediates, perhaps in muscle biopsy tissue of VGWIs in whom muscle is affected, to exclude alterations in CAC markers in VGWIs.

This study should be viewed as hypothesis generating. The sample sizes limit statistical power, particularly for the CAC to PCr-R analysis; but despite this, the strong correlations for five of the seven CAC markers to PCr-R, observed in controls, were significant. Where similar significance is observed in a smaller sample, this requires a greater effect size to be present; the prospect of chance being responsible for significance is equal, *in principle*, with a large or small sample, at the same *p*-value. Nonetheless, replication in a larger sample is clearly desirable, and findings should be viewed as provisional.

## 5. Conclusions

The findings here support a relation of CAC markers to PCr-R in healthy controls. The findings suggest the loss of this relationship in VGWIs. The impaired bioenergetics in VGWIs, indexed by prolonged PCr-R, did not relate to blood concentrations of CAC markers—though shifts in relations among CAC markers were observed. Future studies should seek to assess if findings are replicated in larger samples, reassess this in other patient groups, and seek to investigate the basis of these findings.

## Figures and Tables

**Table 1 ijerph-18-01635-t001:** (a) Participant characteristics: Full sample. (b) Participant characteristics: Subset with post-exercise phosphocreatine recovery time constant (PCr-R) assessed.

(a)		Demographic Characteristics			
		All N = 66	Control N = 33	Case N = 33	Case vs. Control *p*-Value
Age (years), mean (SD)		50.0 (6.98)	49.9 (7.20)	50.2 (6.85)	0.88
Male, %		93.9	93.9	93.9	1.0
Ethnicity, %					
Caucasian		66.7	66.7	66.7	1.0
African American		8.2	8.2	8.2	1.0
Hispanic		9.09	9.09	9.09	1.0
Asian		3.03	3.03	3.03	1.0
Native American		3.03	3.03	3.03	1.0
Married, %		53.0	39.9	66.7	0.027
		**Gulf War Illness (GWI) Kansas Criteria Symptom Scores ***			
	**Rating**	**All** **Mean (SD)**	**Control**	**Case**	**All**
Fatigue	0–12	4.03 (4.33)	0.21 (0.55)	7.85 (2.80)	<0.0001
Pain	0–9	2.83 (3.22)	0.15 (0.44)	5.51 (2.45)	<0.0001
Neuro	0–38	9.52 (11.2)	0.21 (0.55)	18.8 (8.80)	<0.0001
Skin	0–6	1.09 (1.79)	0.03 (0.17)	2.15 (2.03)	<0.0001
Gastrointestinal (GI)	0–8	1.83 (2.46)	0.03 (0.17)	3.64 (2.36)	<0.0001
Respiratory	0–7	1.17 (0.84)	0.03 (0.17)	2.30 (2.04)	<0.0001
**(b)**		**Demographic Characteristics**			
		**All** **N = 14**	**Control** **N = 7**	**Case** **N = 7**	**Case vs. Control *p*-Value**
Age (years), mean (SD)		53.6 (6.37)	53.7 (6.55)	53.6 (6.70)	0.97
Male, %		100	100	100	1.0
Ethnicity, %					
Caucasian		85.7	85.7	85.7	1.0
African American		14.3	14.3	14.3	1.0
Hispanic		0	0	0	N/A
Asian		0	0	0	N/A
Native American		0	0	0	N/A
Married, %		57.1	42.9	71.4	0.28
		**GWI Kansas Criteria Symptom Scores ***			
	**Rating**	**All** **Mean (SD)**	**Control**	**Case**	**Case vs. Control *p*-Value**
Fatigue	0–12	4.36 (4.83)	0.29 (0.49)	8.43 (3.41)	<0.0001
Pain	0–9	2.93 (3.10)	0.29 (0.49)	5.57 (2.07)	<0.0001
Neuro	0–33	10.1 (11.7)	0.29 (0.76)	20.0 (8.45)	<0.0001
Skin	0–6	1.14 (1.92)	0	2.29 (2.21)	0.0182
GI	0–7	1.86 (2.51)	0	3.71 (2.36)	0.0013
Respiratory	0–5	1.64 (1.74)	0.14 (0.38)	3.14 (1.07)	<0.0001

* GWI cases are selected for having Kansas symptoms above a threshold; controls are selected for having Kansas symptom scores below a threshold. This is a result of cases being selected for adherence to Kansas symptom inclusion criteria, and controls for failing to meet Kansas symptom inclusion criteria.

**Table 2 ijerph-18-01635-t002:** (a) Citric Acid Cycle (CAC) markers (μmol/L): Full sample. (b) CAC markers (μmol/L): PCr-R subset.

(a)					
CAC Marker	N	All N = 66 Mean (SD)	Control N = 33	Case N = 33	Case vs. Control *p*-Value *
succinate	66	4.13 (1.39)	4.21 (1.53)	4.04 (1.24)	0.61
malate	66	6.10 (2.26)	6.24 (2.19)	5.95 (2.35)	0.47
fumarate	66	1.46 (0.63)	1.44 (0.56)	1.48 (0.70)	0.79
alpha ketoglutarate	66	12.7 (8.78)	12.7 (10.9)	12.7 (6.1)	0.99
aconitate	66	4.15 (0.84)	4.09 (0.74)	4.22 (0.93)	0.42
citrate	62	14.1 (4.34)	14.8 (4.27)	13.3 (4.36)	0.199
isocitrate	62	5.44 (2.38)	5.70 (2.21)	5.17 (2.54)	0.37
**(b)**					
**CAC Marker**		**All** **N = 14** **Mean (SD)**	**Control** **N = 7**	**Case** **N = 7**	**Case vs. Control *p*-Value**
succinate		3.76 (1.23)	3.62 (1.32)	3.90 (1.21)	0.69
malate		5.41 (1.68)	5.73 (2.02)	5.11 (1.34)	0.51
fumarate		1.25 (0.35)	1.15 (0.38)	1.35 (0.31)	0.29
alpha ketoglutarate		9.81 (3.80)	8.19 (2.17)	11.4 (4.52)	0.11
aconitate		4.13 (0.57)	4.00 (0.48)	4.26 (0.66)	0.42
citrate		13.4 (3.71)	12.9 (4.83)	14.0 (2.49)	0.64
isocitrate		5.13 (2.31)	4.26 (2.44)	6.01 (1.98)	0.20

* Paired *t*-test.

**Table 3 ijerph-18-01635-t003:** (a) Controls: Correlations among CAC markers. (b) Cases: Correlations among CAC markers.

(a)								
	Succinate	Fumarate	Malate	AKG	Aconitate	Citrate		Isocitrate
Succinate		0.54	0.74	0.62	0.37	0.56		0.45
		0.0011	<0.0001	0.0001	0.035	0.0010		0.010
Fumarate	0.54		0.67	0.33	0.049	0.68		0.70
	0.0011		<0.0001	0.064	0.79	<0.0001		<0.0001
Malate	0.74	0.67		0.29	0.56	0.76		0.58
	<0.0001	<0.0001		0.10	0.0008	<0.0001		0.0006
Alpha ketoglutarate (AKG)	0.62	0.33	0.29		0.18	0.070		0.18
	0.0001	0.064	0.10		0.33	0.71		0.32
Aconitate	0.37	0.049	0.56	0.18		0.40		0.18
	0.035	0.79	0.0008	0.33		0.027		0.32
Citrate	0.56	0.68	0.76	0.070	0.40			0.74
	0.0010	<0.0001	<0.0001	0.71	0.027			<0.0001
Isocitrate	0.45	0.70	0.58	0.18	0.18	0.74		
	0.010	<0.0001	0.0006	0.32	0.32	<0.0001		
*PCr-R*	−0.76	−0.85	−0.74	−0.41	−0.34	−0.88		−0.81
	0.048	0.014	0.058	0.36	0.45	0.022		0.052
**(b)**								
	**Succinate**	**Fumarate**	**Malate**	**AKG**	**Aconitate**	**Citrate**	**Isocitrate**	
Succinate		0.67	0.52	0.56	0.42	0.39	0.31	
		<0.0001	0.0018	0.0006	0.014	0.031	0.086	
Fumarate	0.67		0.48	0.42	0.17	0.23	0.25	
	<0.0001		0.0043	0.016	0.35	0.22	0.18	
Malate	0.52	0.48		0.62	0.32	0.59	0.66	
	0.0018	0.0043		0.0001	0.071	0.0005	0.0001	
AKG	0.56	0.42	0.62		0.32	0.26	0.58	
	0.0006	0.016	0.0001		0.071	0.15	0.0006	
Aconitate	0.42	0.17	0.26	0.32		0.26	0.086	
	0.014	0.35	0.15	0.071		0.15	0.65	
Citrate	0.39	0.23	0.59	0.26	0.26		0.70	
	0.031	0.22	0.0005	0.15	0.15		<0.0001	
Isocitrate	0.31	0.25	0.66	0.58	0.086	0.70		
	0.086	0.18	0.0001	0.0006	0.65	<0.0001		

For each row variable, the first sub-row represents the correlation coefficients, and the lower sub-row represents the *p*-values of the correlations.

**Table 4 ijerph-18-01635-t004:** AKG and aconitate Relationships to other CAC intermediates (*p*-values).

	AKG Control	AKG Case		Aconitate Control	Aconitate Case
Succinate	0.014	<0.001	Succinate	0.020	<0.001
Fumarate	0.071	0.222	Fumarate	0.70	0.31
Malate	0.13	0.002	Malate	<0.001	0.014
Aconitate	0.057	0.076	AKG	0.017	0.007
Citrate	0.21	0.13	Citrate	0.051	0.15
Isocitrate	0.030	0.005	Isocitrate	0.354	0.43

*p*-values are from regression with robust SE, with the column headers as dependent variables, and the row headers as independent variables, in cases and controls—adjusted for archive time.

**Table 5 ijerph-18-01635-t005:** Relation of PCr-R to CAC markers: Correlation and regression with robust standard errors

	(a) PCr-R Relates Strongly to Many CAC Intermediates in Controls						
	Controls						
	Correlation of CAC Marker with PCr-R						
	Succinate	Fumarate	Malate	Citrate	Isocitrate	AKG	Aconitate
r	−0.76	−0.85	−0.74	−0.88	−0.81	−0.41	−0.34
*p*	0.048	0.014	0.058	0.022	0.052	0.36	0.45
	**Prediction of CAC Marker by PCr-R: Regression with Robust SE**						
	**Succinate**	**Fumarate**	**Malate**	**Citrate**	**Isocitrate**	**AKG**	**Aconitate**
β (SE)	−0.12 (0.035)	−0.037 (0.0076)	−0.17 (0.053)	−0.46 (0.11)	−0.22 (0.036)	−0.15 (0.081)	−0.0045 (0.020)
*p*	0.021	0.005	0.022	0.014	0.004	0.13	0.84
	**(b) PCr-R Does Not Relate to Any CAC Intermediates in Cases**						
	**Cases**						
	**Correlation of CAC Marker with PCr-R**						
	**Succinate**	**Fumarate**	**Malate**	**Citrate**	**Isocitrate**	**AKG**	**Aconitate**
r	0.29	−0.27	0.25	−0.18	−0.22	0.18	0.18
*p*	0.52	0.56	0.58	0.73	0.67	0.70	0.70
	**Prediction of CAC Marker by PCr-R: Regression with Robust SE**						
	**Succinate**	**Fumarate**	**Malate**	**Citrate**	**Isocitrate**	**AKG**	**Aconitate**
β (SE)	0.020 (0.025)	−0.0047 (0.0078)	0.019 (0.039)	−0.023 (0.051)	−0.022 (0.046)	0.046 (0.11)	+0.017 (0.014)
*p*	0.47	0.57	0.65	0.67	0.65	0.69	0.29

In regression analyses, aconitate and AKG values are adjusted for archive time. *p*-values, for both cases and controls, are nonsignificant with or without this adjustment.

## Data Availability

The data presented in this study are available on request from the corresponding author. The data are not publicly available because we are seeking to protect participant confidentiality and health information.

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
