# Peer review of "A Pilot Study of Bioenergetic Marker Relationships in Gulf War Illness: Phosphocreatine Recovery vs. Citric Acid Cycle Intermediates"

_ijerph, 2021, doi:10.3390/ijerph18041635_

Round 1

Reviewer 1 Report

The authors measure citric acid cycle intermediates in blood and phosphocreatine recovery in muscle from patients with Gulf War Illness (GWI) and controls. The conclusion is that there are not statistical differences in blood citric acid cycle intermediates in veterans with GWI compared to controls. However in controls there were correlations between citric acid cycle intermediate levels that were lost in veterans with GWI.  The study was well-designed and well-written. The statistical analysis appears appropriate.  However, there are a few issues where parts are unclear.

Comment 1: In Table 1A there appears to be two typos including:

African American -All  18.2 -> 8.2

Native American – All   3.09 -> 3.03

Comment 2: The authors should mention in the text that there are more married individuals in your VGWI cohort for citric acid cycle intermediate measurements than in your control cohort as this may affect the results.

Comment 3: In Table 3c it is unclear what is being stated about archival time of the samples.  Please be more specific on which citric acid cycle intermediates are degrading over time and how the data were adjusted to compensate. Were the samples kept in the freezer? Can you suggest a way to prevent this sample degradation from occurring in the future? You stated that “Since archive time appeared to affect these 167 variables, relationships were evaluated adjusting for the time variable.”   Are there any other examples in the literature of citric acid cycle intermediates degrading over time in blood samples while being stored? Please clarify.

Comment 4: It is not that surprising that citric acid cycle intermediate levels in blood samples were not affected when GWI is more of a neuromuscular disorder.  For future studies, are there protocols currently available to measure citric acid cycle intermediates in brain or muscle of live human subjects? Please discuss. One group appeared to be able to be measured in live rodent brains. J Cereb Blood Flow Metab. 2012 Dec; 32(12): 2108–2113. doi: 10.1038/jcbfm.2012.136. PMCID: PMC3519415. In vivo detection of brain Krebs cycle intermediate by hyperpolarized magnetic resonance. Such a 13C NMR technique will likely be needed to make progress in this field to better diagnose and treat mitochondrial issues in the affected tissues.

Reviewer 2 Report

  1. Title: Change the title to add pilot study as not enough power for anything else 
  2. Line 37 - add the White cortex paper as reference -
  3. Line 45 - need a link between exposures and health symptoms/ bioenergetic impairment if you are going to discuss this, especially in relationship to GWI criteria -  
  4. Line 60 - CAC - link this to prior paragraph, it isn't clear what you are looking for and how it relates to the measures, which part of the cycle is expected to be related - 
  5. Line 65- this is unclear -
  6. Line 83 - the control group is insufficient as to of course GWI veterans would differ from "normals", what about GWI versus non-ill or to look at exposure type - 
  7. Analyses could have been conducted as to specific health symptoms, rather than GWI versus normals, as in those with fatigue, or muscle ache. - Doing a sub anlaysis of the kansas symptom domains. Is bioenergetic impairment related to all symptom domains or just fatigue? 
Specifically:   Introduction: What is bioenergetic impairment and how does it related to the symptoms of Gulf War Illness? Perhaps a discussion of fatigue is warranted.    Materials and Methods: Was Kansas exclusionary criteria not applied to the veterans? I see you describe it for the controls. If it wasn't used for the veterans, please state the reason, and then include it in the limitations section.    Where were the healthy non-veterans recruited from? Partnering study?    What about a veteran comparison group? Such as GW controls, or non-deployed group? Was PTSD taken into account? Does PTSD affect CAC? If so, what was the rate of current PTSD in the GW veterans?    Lines 105-106: was a sub analysis done removing the pairing that were not shared across the two sets? This is quite confusing. Why weren't the same pairs used?    A more detailed description of what "archive time" is and why it is important to control for is warranted.     The PCr-R analyses were done after an exercise challenge, but the CAC marker analyses were not. Could that influence the findings? The authors should describe why the CAC markers were also not taken after an exercise challenge, and should include this in the limitations section. 

Reviewer 3 Report

Comments and Suggestions for Authors

  • Statistical data must be provided to support the abstract results.
  • In the introduction section, the authors should formulate the hypothesis of the study
  • The authors should specify which is the study population, from which the cases and controls are obtained.
  • The authors should specify how the study population was accessed and what was the procedure used
  • The authors have not indicated whether the study has a favorable report from the corresponding institutional Ethics Committee. Specify report number and date
  • The authors have not indicated whether the participants had to sign the informed consent before the start of the study.
  • The authors should specify how they have evaluated the severity of the symptoms in the cases
  • The authors should explain why the controls are obtained among non-veterans. The logical thing would have been to do it among veterans who did not attend the Gulf War
  • The authors should better explain the following sentence "a half match for ethnicity was qyalifying"
  • In the material and methods section, the authors should also specify that the controls were selected from those people who had values ​​below a threshold in the Kansas symptoms score.
  • The authors should specify which are the thresholds of the Kansas symtoms score
  • The authors should explain which are the secondary variables of the study
  • The authors should specify under which conditions the blood collection was carried out from the participants. Was it necessary to perform more than one blood collection on some of the participants?
  • The authors should specify the type, characteristics and conditions of the exercise that the participants had to perform.
  • What is the reason why PCr-R was only assessed in 14 participants, and not in the entire sample? It has not been explained throughout the study
  • Regarding the sample, despite the fact that GWI affects more women, most of the participants are men. Explain the reason and indicate it as a limitation of the study
  • The sample size is very small, which prevents the possible generalization of the results. Was this size calculated in any way?
  • If GWI cases are selected for having Kansas symptoms above a threshold, obviously the relationship between cases and controls is going to be significant.
  • Table 1 and Figure 1 represent the same, so the authors should eliminate one of them
  • As for the discussion section, the first part should be included in results. This section should begin with a brief summary of the results obtained.
  • The discussion section hardly compares with the results obtained in other studies. This section should be expanded

Round 2

Reviewer 3 Report

Accept in present form